# Fed-Batch Sucrose Crystallization Model for the B Massecuite Vacuum Pan, Solution by Deterministic and Heuristic Methods

**Paulo Eduardo de Morais Gonzales \***, **Marcos Antônio de Souza Peloso, Jr., José Eduardo Olivo and Cid Marcos Gonçalves Andrade**

Chemical Engineering Department, State University of Maringá, Colombo Av. 5790, Maringá 87020-900, Brazil; marcosspeloso@gmail.com (M.A.d.S.P.J.); jeolivo@uem.br (J.E.O.); cmgandrade@uem.br (C.M.G.A.)

**\*** Correspondence: paulogonzales@outlook.com

**Abstract:** Fed-batch crystallization is a crucial step for sugar production. In order to relate parameters that are difficult to measure (average diameter of the crystals and total mass formed) to other easier to measure parameters (volume, temperature, and concentration), a model was developed for a B massecuite vacuum pan composed of mass and energy balances together with empirical relations that describe the crystal development inside equipment. The generated system of ordinary differential equations (ODE) had eight parameters which were adjusted through minimization of relative differences between the model results and experimental data. It was solved through the function fmincon, available in MATLAB$^{TM}$, which is a deterministic and gradient-based optimization method. The objective of this paper is to improve the model obtained and, for this purpose, two metaheuristic functions were used: genetic algorithm and particle swarm. To compare the results, the convergence time of each algorithm was used as well as the resulting quadratic deviation. The particle swarm method was the best option among the three used, presenting a shorter execution time and lower quadratic relative deviation.

**Keywords:** sucrose; crystallization; fed-batch; metaheuristic method; deterministic method; modeling

---

## 1. Introduction

Crystallization is the main process in the sugar industry for the separation and obtaining of sucrose in its commercial form, sugar, and is one of the crucial steps to increase the yield from the plant [1]. Among all steps of sugar production, fed-batch crystallization is the most complex in operational terms since it requires greater care by the plant operators [2].

The industrial sucrose crystallization process consists of three basic equipment: vacuum pan, crystallizer, and centrifuge. In the vacuum pan, sugar crystals form and grow, at a constant temperature, in a fed-batch process. In crystallizers, the crystals grow due to the cooling of the massecuite, which increases supersaturation. The centrifuges separate the crystals formed from the mother liquor which, once exhausted, is called molasses. In this sequence, about half of the sucrose is recovered as sugar crystal. To maximize this recovery, crystallization schemes are used, inserting new sequences of this equipment that are fed by the molasses from the previous step. These sequences can be repeated once (two massecuites scheme), or twice (three massecuites scheme). It is interesting to note that because each sequence of these equipment recovers around half of the crystals, a two massecuite scheme recovers 75% of the sucrose as crystal, while a three massecuite scheme recovers around 87.5%. Because the cost of establishing a three massecuites plant is 50% higher with a mere 12.5% increase in sucrose recovery and most Brazilian industries have an ethanol plant attached that consumes molasses,

the two massecuites scheme is the most widespread in the country [2,3]. Figure 1 shows the flowchart of the two massecuites scheme.

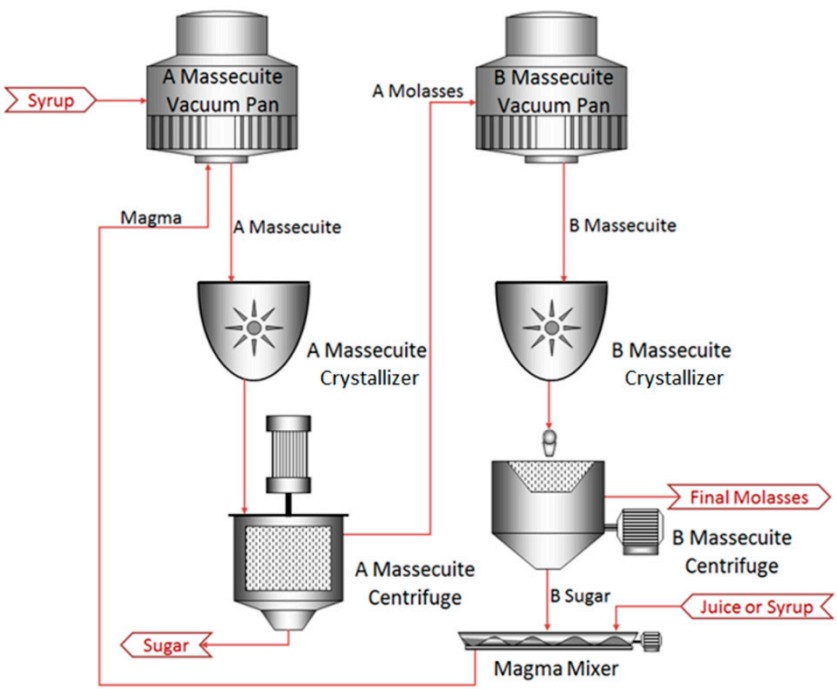

**Figure 1.** Two massecuites scheme flowchart [4], based on [2].

As Brazil is among the largest sugar producers in the world, the scheme of the two massecuites shown in Figure 1 was the one used in the experiments [3]. As local plants also produce electricity with the steam from the boilers, the massecuite cooling in the crystallizers is not used, which eliminates the need to reheat the massecuite to feed the centrifuges (cooling increases the viscosity of the mother liquor) [3,4]. Thus, the crystallization of sucrose occurs only in vacuum pans, making its study the focus of this work.

The instrumentation of massecuite vacuum pans is a challenge for automation companies due to the high values of the sensors and large amount of equipment to be monitored [5,6]. In order to reduce these costs, two models were developed to relate the mean diameter of the crystals D (4,3) and the total formed crystal mass (FCM)—the data of which are difficult to measure—with the equipment concentration, volume, and temperature; one for A massecuite vacuum pan and the other for B massecuite vacuum pan [7]. Both models were solved through the function *fmincon*, available in MATLAB$^{TM}$, which is a deterministic and gradient-based optimization method [8,9].

With these models, it is possible to develop soft sensors to measure crystal size and quantity in the vacuum pan. It is essential that the program is fast and accurate. Thus, to improve the model obtained for B massecuite vacuum pan, which has eight parameters for adjustment (seven kinetic parameters and a thermodynamic data), two metaheuristic methods were used—genetic algorithm (GA) and particle swarm. To compare the results, the convergence time of each algorithm was used as well as the relative resulting quadratic deviation.

## 2. Methodology

The equipment used to obtain the experimental data is installed in a São Paulo state sugarcane industry. During the experiment, the volume, concentration, and temperature data of the fluid were recorded by the sensors installed in the equipment. Each batch occurred in two hours, and samples were taken every fifteen minutes through the equipment probe to evaluate both FCM and D (4,3) through a Nikon Eclipse E200-LED Binocular microscope. Figure 2 shows how the crystals look at the

end of the crystallization process in B massecuite vacuum pan and Figure 3 shows the distribution curve of the crystal size at the last point of the first experiment.

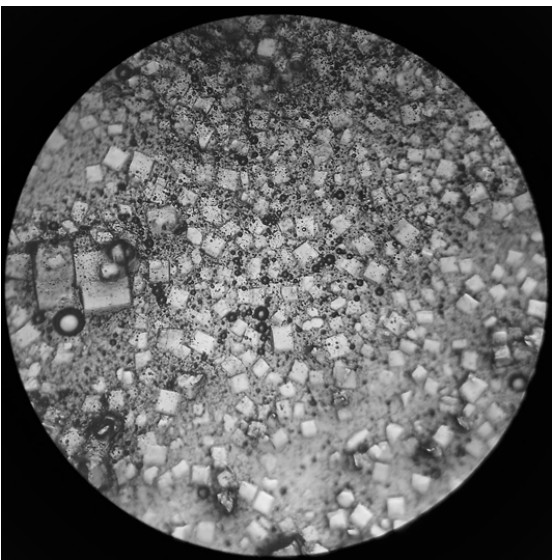

**Figure 2.** Sucrose crystals at the end of the process.

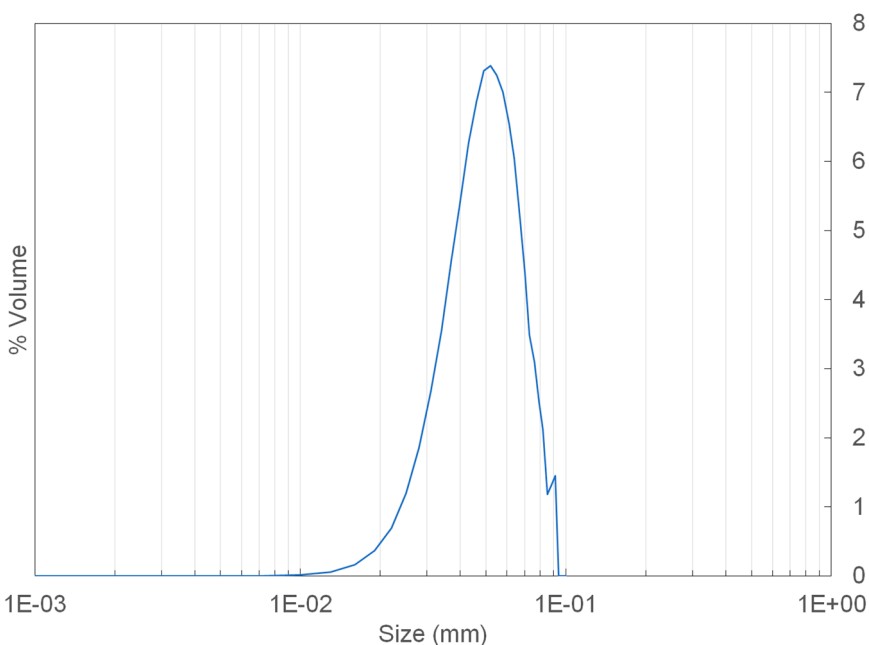

**Figure 3.** Size distribution curve of the last point of the first experiment.

The mathematical modeling was adapted to a model of a process in pilot scale—batch crystallization with cooling (using evaporation and cooling the solution)—as developed in [10]. The pilot scale process starts with approximately 752 cm$^3$ of syrup, operating for 40 min at 60 °C and is then cooled for 50 min to 40 °C. During its operation, the massecuite volume decreases since it is not fed with syrup and due to evaporation of part of its contents. The B massecuite vacuum pan operation starts with one-third of the equipment volume (10 m$^3$) filled with syrup, occurring for two hours at a constant temperature of 80 °C, receiving a variable flow of A molasses so that the volume of equipment reaches its end capacity (30 m$^3$) in compensating for evaporation of the solvent. The differences between the two processes are summarized in Table 1.

**Table 1.** Differences between batch crystallization and B massecuite pan.

| Characteristic | Batch Crystallizer | B Massecuite Vacuum Pan |
|---|---|---|
| Scale | Pilot | Industrial |
| Feed | Just in beginning | Throughout the process |
| Crystals initial size | 0.0019126 cm | 0.030 cm |
| Equipment's fluid volume | Decreases | Increases |
| Operation temperature | 60 °C, followed by cooling to 42 °C | 80 °C throughout the process |

The population balance of crystals of the model used in this paper is calculated using the following partial differential equation (PDE) [7,10,11]. This PDE is shown in Equation (1):

$$\frac{\partial(nV)}{\partial t} + \frac{V\partial(Gn)}{\partial L} = V(\alpha(L)) \tag{1}$$

with initial condition

$$n(L,\ t) = n_0(L,\ t)\ t = 0$$

and the boundary condition

$$n(L_0,\ t) = \frac{B^0}{G_{L=L_0}}. \tag{2}$$

$\alpha(L)$ is known as production-reduction term and it groups the rates of birth and destruction of crystals. Applying the method of moments (MOM) in Equation (1), the following system of ordinary differential equations (ODEs), presented in Equations (3) and (4), is obtained [10]:

$$\frac{d\mu_0}{dt} = B^0 \tag{3}$$

$$\frac{d\mu_k}{dt} = kG\mu_{k-1}(1 + \ln(V)) \quad k = 1\ldots3 \tag{4}$$

where $V$ is the mass volume inside the equipment in cm$^3$. The number of moments necessary for the model resolution can tend to infinity, but due to the use of the empirical equation shown below (Equation (5)), the moment $\mu_3$ enters the mass balance, causing the four moments ($\mu_0$, $\mu_1$, $\mu_2$, and $\mu_3$) to be enough to solve the ODE system [12].

$B^0$ (N° particles/cm$^3$ min) and $G$ (cm/min) are the nucleation and growth rates, respectively, that can be modeled by the empirical power equations presented in Equations (5) and (6) [10]:

$$B^0 = K_b S_r{}^b M_T{}^j N_r{}^p \tag{5}$$

$$G = K_g S_r{}^g N_r{}^q \tag{6}$$

$N_r$ is the stirrer speed in rpm. The terms $K_b$ (N° part/cm$^3$·min·(g/cm$^3$)$^j$·(rpm)$^p$), $b$, $j$, $p$, $K_g$ (cm/min·(rpm)$^q$), $g$, and $q$ are the kinetic parameters of these equations and were estimated and optimized using the experimental data.

In Equation (5), $M_T$ represents the total mass of crystals and can be obtained using Equation (7):

$$M_T = \rho_c K_v \mu_3(t) \tag{7}$$

where $\rho_c$ is the density of the sucrose in g/cm$^3$, and $K_v = \pi/6$ is the characteristic form factor for sugar crystals [10,13].

In Equations (5) and (6), $S_r$ represents the relative supersaturation, and is defined by Equation (8):

$$S_r = \frac{C - C_{sat}}{C_{sat}} \tag{8}$$

where $C$ is the sucrose concentration and $C_{sat}$ is the saturation concentration point, both in g/cm$^3$. The mass percentage of sucrose in the saturated solution ($Brix_{sat}$) can be calculated as a function of temperature using Equation (9) [4,7,14].

$$Brix_{sat} = 5.1844 \times 10^{-4}T^2 + 1.3575 \times 10^{-1}T + 64.168 \tag{9}$$

where $T$ is the temperature in °C.

The density of a sucrose solution ($\rho$) can be obtained by Equation (10) ($T$ in °C) [7,15]:

$$\rho = \left(1 + Brix\frac{(Brix + 200)}{54000}\right) \times \left(1 - 0.036\frac{(T - 20)}{(160 - T)}\right). \tag{10}$$

The solution concentration, as well as its saturation concentration, can be obtained by Equation (11):

$$C = \frac{\rho \times Brix}{100}. \tag{11}$$

The mass balance of the process is presented in Equation (12) [7]:

$$m_{sucrose}(t) = C_{mA} V_{mA} + m_{sucrose^0} \tag{12}$$

where $m_{sucrose}$ and $m_{sucrose}{}^0$ is the total mass of sucrose (g) in the equipment at time t and 0, respectively, $C_{mA}$ is the concentration of A-molasses (g/cm$^3$), and $V_{mA}$ is the accumulated flow of A-molasses added in equipment (cm$^3$). Equation (13) shows that the total mass of sucrose inside the equipment is partly in solution ($CV_S$) and partly crystallized ($\rho_c V_c$):

$$m_{sucrose}(t) = C\,V_S + \rho_c V_c. \tag{13}$$

Equation (14) shows that the volume of the solution ($V_S$) plus the volume of the crystals ($V_c$) is equivalent to the volume of massecuite in the equipment ($V$).

$$V = V_S + V_c \tag{14}$$

Rearranging:

$$V_S = V - V_c. \tag{15}$$

Thus, replacing Equations (13) and (14) in Equation (12):

$$C\,(V - V_c) + \rho_c V_c = C_{mA} V_{mA} + m_{sucrose^0}, \tag{16}$$

deriving Equation (16) the Equation (17) is obtained:

$$\frac{dC}{dt}\,(V - V_c) + C\left(\frac{dV}{dt} - \frac{dV_c}{dt}\right) + \rho_c\frac{dV_c}{dt} = C_{mA}\frac{dV_{mA}}{dt}. \tag{17}$$

Substituting $V_c$ for $K_v\mu_3(t)$ and isolating $dC/dt$ leads to Equation (18):

$$\frac{dC}{dt} = \left(\frac{1}{V - \mu_3(t)K_v}\right) \times \left(C_{mA}\frac{dV_{mA}}{dt} + (C - \rho_c)K_v\frac{d\mu_3}{dt} - C\frac{dV}{dt}\right) \tag{18}$$

with initial condition

$$C(0) = C_0. \tag{19}$$

The energy balance of B massecuite vacuum pan is shown in Equation (20) [2]:

$$E(t) = Q - W + F_{in}h_{in} - F_{out}h_{out} \tag{20}$$

where $E(t)$ is the internal energy of the system in J, $F_{in}$ and $F_{out}$ are the mass input and output in g, $h_{in}$ and $h_{out}$ the relative enthalpies in J/g. Replacing the Equation (20) variables with those of the B vacuum pan [7,10]:

$$m_{VC}Cp_{VC}\Delta T_{VC} = Q - W + m_{mA}Cp_{mA}\Delta T_{mA} - m_{ev}H_{ev} - m_c\Delta H_c. \tag{21}$$

The subscripts $_{mA}$ and $_{VC}$ represent A molasses and mass in vacuum pan, respectively. Deriving Equation (21):

$$m_{VC}Cp_{VC}\frac{dT}{dt} = \dot{Q} - \dot{W} + Cp_{mA}\Delta T_{mA}\frac{dm_{mA}}{dt} - \frac{d(m_{ev}H_{ev})}{dt} - \Delta H_c\frac{dm_c}{dt}. \tag{22}$$

The heat rate can be obtained by Equation (23) [2]:

$$\dot{Q} = U_l A_l \left(T_j - T\right). \tag{23}$$

$T_j$ is the temperature of the steam in the equipment's calandria (°C), $U_l$ is the global coefficient of thermal exchange (J/(°C·min·cm$^2$)), and $A_l$ is the heat exchanger area (cm$^2$). The work rate is calculated by Equation (24) [11]:

$$\dot{W} = -\frac{d(PV)}{dt} \tag{24}$$

where $P$ is the absolute pressure in J/cm$^3$. Replacing Equations (23) and (24) in Equation (22) and explaining the volumes of A molasses and crystals formed, Equation (25) is obtained:

$$\frac{dT}{dt} = \frac{\frac{d(PV)}{dt} - \frac{d(H_{ev}m_{ev})}{dt} + (\rho Cp T)_{mA}\frac{dV_{mA}}{dt} - \Delta H_c\rho_c K_v\frac{d\mu_3}{dt} + U_l A_l\left(T_j - T\right)}{(\rho V Cp)_{VC}} \tag{25}$$

with initial condition

$$T(0) = T_0. \tag{26}$$

$\Delta H_c$ is the crystallization heat in J/g, which can be calculated by means of Equation (27) ($T$ in °C) [10,13]:

$$\Delta H_c = -12.2115 - 0.7937T. \tag{27}$$

Also in Equation (25), $m_{ev}$ is the evaporated accumulated mass flow rate and $H_{ev}$ is the vaporization enthalpy in J/g, which can be obtained by Equation (28) ($T$ in °C) [16]:

$$H_{ev} = 4.1868 \times (607 - 0.7T). \tag{28}$$

The absolute pressure $P$, in J/cm$^3$, within the equipment can be obtained using the Antoine equation presented in Equation (29) ($T$ in °C) [16]:

$$P = 2.21 \times 10^1 e^{\left(6.53247 - \frac{7173.79}{1.8T + 421.4747}\right)}. \tag{29}$$

The specific heat of sucrose solutions, $Cp$ in J/g °C, can be obtained by means of Equation (30) ($T$ in °C) [7,15,17]:

$$Cp = p_1 + p_2.Brix + p_3.Brix.T + p_4.T + p_5.T^2. \tag{30}$$

The dynamics of the volume of liquid inside the equipment, $V$ in cm$^3$, is described by Equation (31), which is a cubic correlation obtained through the equipment data during the experiment ($t$ in min):

$$V(t) = -1.5896t^3 - 75.299t^2 + 1.9859 \times 10^5 t + 10^7. \tag{31}$$

Table 2 presents the parameters used in this modeling:

**Table 2.** Process parameters to solve the population, mass, and energy balances.

| Parameter | Value | Unit |
|:---:|:---:|:---:|
| $K_v$ | $\pi/6$ | – |
| $A_l$ | 1950000 | cm$^2$ |
| $L_0$ | 0.030 | cm |
| $p_1$ | 4.12553 | J/g °C |
| $p_2$ | −0.024804 | J/g °C |
| $p_3$ | 0.000067 | J/g (°C)$^2$ |
| $p_4$ | 0.0018691 | J/g (°C)$^2$ |
| $p_5$ | −0.000009271 | J/g (°C)$^3$ |
| $T_{mA}$ | 80 | °C |
| $T_0$ | 80 | °C |
| $T_j$ | 131 | °C |
| $C_{mA}$ | 1.0085 | g/cm$^3$ |
| $C_0$ | 1.0733 | g/cm$^3$ |
| $N_r$ | 300 | rpm |

In Equations (5) and (6), the proposed mathematical model includes seven kinetic parameters ($K_b$, $b$, $j$, $p$, $K_g$, $g$, and $q$) specific for each batch crystallization operating condition [10]. These values need to be adjusted so that the model provides representative results of this process. The global coefficient of thermal exchange ($U_I$) was also added in the adjustment [7]. Based on this, it sought to reduce the relative quadratic differences between the experimental data and those obtained by the model, according to Equation (32):

$$min \ \sum_{i=1}^{n} \left( \frac{CSD_{exp,i} - CSD_{model,i}}{CSD_{exp,i}} \right)^2 \tag{32}$$

In the model deterministic solution, the non-linear optimization tool *fmincon*, available in MATLAB$^{\text{TM}}$, was used in conjunction with the use of the *ode23s* [7,10]. In this paper Equation (32) was also solved using two metaheuristic tools, one based on the genetic algorithm (*ga*) and the other on the particle swarm (*particleswarm*) [18,19]. The results were obtained with the hardware configuration shown in Table 3:

**Table 3.** Hardware configuration used in the modeling.

| | |
|:---:|:---:|
| **CPU** | i7 4770k |
| **RAM** | 2 × DDR3 1600 MHz-08 GB |
| **HD** | 240 GB SSD SATA 3 |

The algorithms were executed in order to record the convergence time and the relative quadratic deviation of the obtained results with the experimental data.

## 3. Results and Discussion

Table 4 presents the operational data from experiments 1 and 2:

**Table 4.** Operational data from experiments 1 and 2 ($V_{mA}$ is accumulative).

| | Experiment 1 | | | | | Experiment 2 | | | | |
|---|---|---|---|---|---|---|---|---|---|---|
| t (min) | $V_{VC}$ (m³) | $T$ (°C) | $C$ (g/cm³) | $S_r$ | $V_{mA}$ (m³) | $V_{VC}$ (m³) | $T$ (°C) | $C$ (g/cm³) | $S_r$ | $V_{mA}$ (m³) |
| 0 | 10.02 | 78.98 | 1.072 | 0.018 | 0.000 | 9.87 | 80.81 | 1.074 | 0.003 | 0.000 |
| 5 | 11.17 | 80.44 | 1.079 | 0.022 | 1.420 | 10.64 | 80.13 | 1.077 | 0.021 | 1.900 |
| 10 | 12.25 | 80.04 | 1.079 | 0.027 | 2.495 | 11.74 | 79.62 | 1.079 | 0.035 | 3.905 |
| 15 | 13.23 | 80.53 | 1.084 | 0.038 | 3.693 | 13.28 | 79.79 | 1.089 | 0.063 | 4.686 |
| 20 | 13.93 | 79.13 | 1.085 | 0.061 | 4.681 | 13.77 | 80.37 | 1.086 | 0.047 | 5.714 |
| 25 | 15.05 | 80.45 | 1.088 | 0.053 | 6.058 | 13.83 | 81.10 | 1.089 | 0.046 | 6.680 |
| 30 | 15.96 | 79.79 | 1.090 | 0.068 | 7.159 | 14.80 | 80.40 | 1.090 | 0.059 | 8.022 |
| 35 | 16.87 | 80.22 | 1.091 | 0.064 | 8.008 | 15.91 | 79.85 | 1.091 | 0.072 | 9.205 |
| 40 | 17.58 | 79.00 | 1.093 | 0.091 | 9.771 | 16.86 | 80.00 | 1.091 | 0.070 | 11.123 |
| 45 | 18.09 | 80.27 | 1.094 | 0.077 | 11.705 | 17.74 | 80.24 | 1.093 | 0.071 | 12.484 |
| 50 | 19.27 | 78.81 | 1.094 | 0.096 | 13.382 | 19.43 | 79.84 | 1.094 | 0.083 | 13.872 |
| 55 | 21.04 | 80.16 | 1.094 | 0.076 | 14.258 | 21.03 | 79.13 | 1.095 | 0.096 | 14.780 |
| 60 | 21.76 | 80.00 | 1.095 | 0.084 | 16.005 | 22.67 | 79.58 | 1.096 | 0.092 | 15.652 |
| 65 | 22.33 | 80.03 | 1.098 | 0.093 | 17.458 | 23.88 | 79.61 | 1.095 | 0.088 | 16.757 |
| 70 | 22.79 | 80.43 | 1.096 | 0.082 | 18.858 | 24.20 | 79.40 | 1.097 | 0.100 | 17.418 |
| 75 | 24.18 | 79.80 | 1.098 | 0.095 | 20.022 | 24.85 | 79.70 | 1.099 | 0.102 | 17.997 |
| 80 | 25.77 | 79.80 | 1.101 | 0.107 | 21.300 | 24.81 | 80.03 | 1.099 | 0.095 | 19.391 |
| 85 | 25.71 | 80.10 | 1.102 | 0.106 | 22.991 | 25.82 | 79.46 | 1.101 | 0.112 | 21.282 |
| 90 | 26.34 | 80.70 | 1.104 | 0.104 | 24.296 | 27.40 | 79.83 | 1.106 | 0.125 | 23.182 |
| 95 | 26.33 | 79.77 | 1.108 | 0.135 | 25.278 | 27.65 | 80.49 | 1.107 | 0.119 | 24.570 |
| 100 | 26.60 | 79.64 | 1.111 | 0.149 | 26.030 | 28.42 | 80.28 | 1.108 | 0.127 | 25.223 |
| 105 | 27.54 | 79.28 | 1.113 | 0.160 | 27.339 | 28.89 | 80.25 | 1.115 | 0.154 | 26.433 |
| 110 | 29.20 | 80.80 | 1.118 | 0.158 | 28.310 | 28.80 | 79.91 | 1.120 | 0.180 | 28.144 |
| 115 | 29.60 | 80.36 | 1.126 | 0.197 | 29.465 | 30.12 | 80.28 | 1.125 | 0.194 | 29.655 |
| 120 | 30.23 | 80.21 | 1.134 | 0.232 | 30.086 | 30.87 | 79.60 | 1.134 | 0.241 | 30.394 |

In these conditions, the size of formed crystals and their quantity were evaluated, shown in Table 5:

**Table 5.** B Massecuite crystals experimental data.

| | Experimental 1 | | Experimental 2 | |
|---|---|---|---|---|
| t (min) | D (4,3) (cm) | MCF (ton) | D (4,3) (cm) | MCF (ton) |
| 0 | 0.029 | 3.68 | 0.030 | 3.83 |
| 15 | 0.034 | 5.86 | 0.034 | 5.86 |
| 30 | 0.034 | 6.04 | 0.036 | 6.95 |
| 45 | 0.038 | 8.46 | 0.039 | 9.07 |
| 60 | 0.044 | 12.26 | 0.042 | 10.98 |
| 75 | 0.044 | 12.62 | 0.043 | 11.53 |
| 90 | 0.047 | 15.34 | 0.047 | 15.34 |
| 105 | 0.053 | 21.49 | 0.049 | 17.80 |
| 120 | 0.054 | 23.60 | 0.056 | 25.40 |

With known data from Tables 2, 4 and 5, the ODE system was solved using the two meta-heuristic methods to reduce the differences between the results of the model obtained in relation to the two experiments (Equation (32)), thereby, obtaining the unknown parameters—the seven kinetic parameters and global coefficient of thermal change ($U_l$) of system—presented in Tables 6 and 7 along

with the parameters of deterministic solution. Although they have different values, the order of magnitude remained.

**Table 6.** Kinetic parameters of crystals nucleation rate.

| Solution | $K_b$ (N° Part/cm$^3$·min·(g/cm$^3$)$^j$·(rpm)$^p$) | $b$ | $j$ | $p$ |
|---|---|---|---|---|
| Deterministic | $9.91 \times 10^{-3}$ | $8.00 \times 10^{-4}$ | $4.00 \times 10^{-3}$ | 1.16 |
| GA | $9.10 \times 10^{-3}$ | $6.00 \times 10^{-4}$ | $3.53 \times 10^{-3}$ | 1.16 |
| Particle Swarm | $9.00 \times 10^{-3}$ | $5.11 \times 10^{-4}$ | $7.99 \times 10^{-3}$ | 0.90 |

**Table 7.** Kinetic parameters of crystals growth rate and global coefficient of thermal change.

| Solution | $K_g$ (cm/min·(rpm)$^q$) | $g$ | $Q$ | $U_l$ (J/(°C min cm$^2$)) |
|---|---|---|---|---|
| Deterministic | $1.89 \times 10^{-6}$ | $1.50 \times 10^{-1}$ | $4.55 \times 10^{-1}$ | 0.8745 |
| GA | $4.35 \times 10^{-6}$ | $1.51 \times 10^{-1}$ | $3.09 \times 10^{-1}$ | 0.8770 |
| Particle Swarm | $3.81 \times 10^{-6}$ | $0.90 \times 10^{-1}$ | $2.89 \times 10^{-1}$ | 0.8720 |

Table 8 presents the average execution time of each algorithm and the relative quadratic deviation obtained at the end of the execution of the algorithm.

**Table 8.** Quadratic deviations and the execution time of each algorithm.

| Solution | Deviation$^2$ | Time (min) |
|---|---|---|
| Deterministic | 1.58% | 7.12 |
| GA | 0.87% | 4.27 |
| Particle Swarm | 0.77% | 2.82 |

It is noted that the particle swarm method was able to obtain a smaller deviation with a shorter execution time compared to the other methods. The genetic algorithm also performed better than the deterministic method used by the *fmincon* function. It is noteworthy that some point results of the deterministic solution were able to converge in less time than the average of the other methods, but without approaching the deviations obtained by them. This occurs due to the high dependence of the deterministic methods of the initial point, which in most cases leads them to local minimums [20].

In the graphs shown in Figures 4–6, it is possible to verify that all the algorithms made a good adjustment to the data.

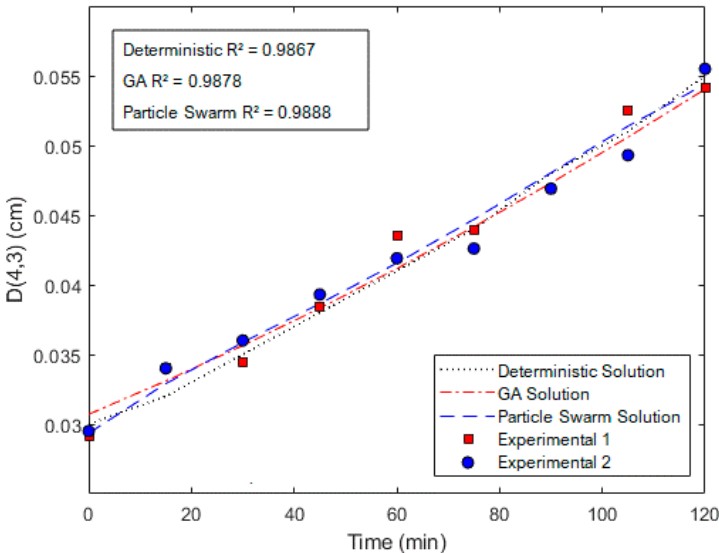

**Figure 4.** Model results—average crystal size D (4,3).

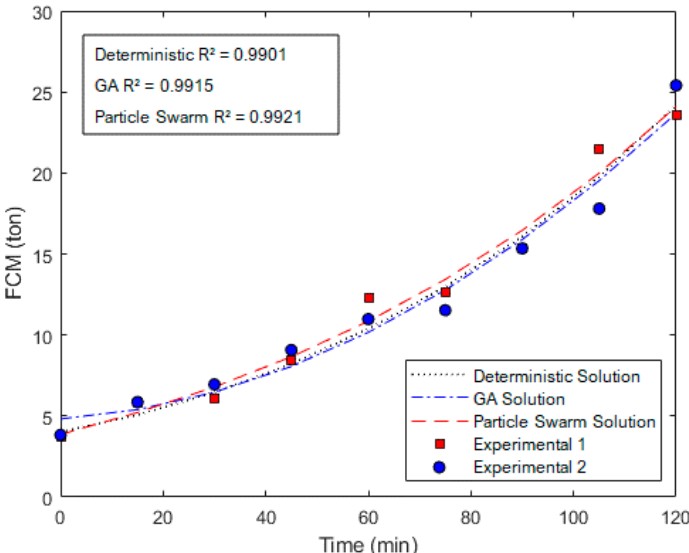

**Figure 5.** Model results—formed crystal mass (FCM).

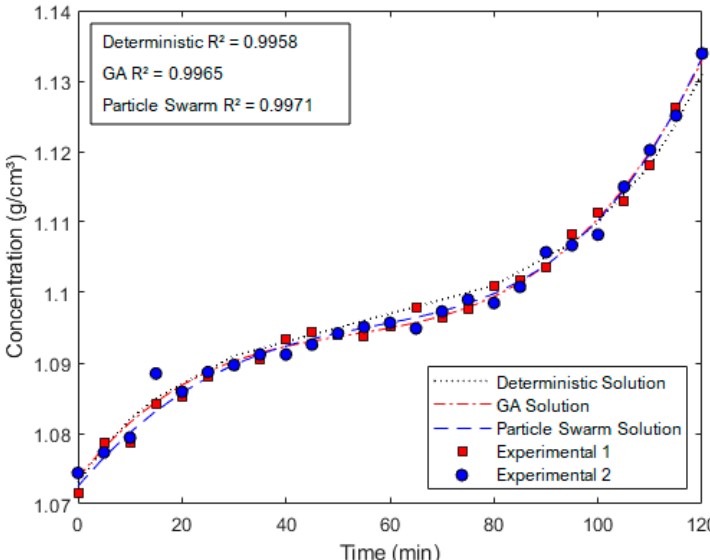

**Figure 6.** Model results—concentration g/cm$^3$.

The correlation coefficient ($R^2$) values were obtained through multiple linear regressions with both experiments, and followed the behavior of the relative square deviations shown in Table 8. The graphs obtained for the metaheuristic methods were similar to those obtained for the deterministic method [7].

## 4. Conclusions

This paper has tried to refine a model obtained for B massecuite vacuum pan by implementing metaheuristic methods in the optimization of the kinetic parameters and global coefficient of thermal change. As shown in Table 8 and Figures 4–6, the particle swarm method was the best option among the three used, presenting a shorter execution time, a lower quadratic relative deviation, and higher correlation coefficients ($R^2$). Nevertheless, when analyzing the data fit, it is noted that all solutions were very close, making the execution time the determining variable for choosing the algorithm to be used, keeping in mind that the development of a soft sensor to measure the size of crystals and their quantity in the vacuum pan is the target of this work. The solution through the deterministic method was highly influenced by the initial point, as expected. As a next step to this work, the refinement of

the model of the operation for A massecuite vacuum pan can be carried out, as well as the use of other functions for comparison.

**Author Contributions:** Conceptualization, P.E.d.M.G.; methodology, P.E.d.M.G.; software, P.E.d.M.G.; validation, P.E.d.M.G., M.A.d.S.P.J. and J.E.O.; formal analysis, P.E.d.M.G.; investigation, P.E.d.M.G. and M.A.d.S.P.J.; resources, P.E.d.M.G., M.A.d.S.P.J. and J.E.O.; data curation, P.E.d.M.G.; writing—original draft preparation, P.E.d.M.G.; writing—review and editing, P.E.d.M.G. and M.A.d.S.P.J.; visualization, P.E.d.M.G. and C.M.G.A.; supervision, C.M.G.A.; project administration, J.E.O. and C.M.G.A.; funding acquisition, M.A.d.S.P.J. and C.M.G.A. All authors have read and agreed to the published version of the manuscript.

**Funding:** This research was funded by National Council for Scientific and Technological Development (CNPq) and by Coordination for the Improvement of Higher Education Personnel (CAPES).

**Acknowledgments:** The authors would like to thank the National Council for Scientific and Technological Development (CNPq) and the Coordination for the Improvement of Higher Education Personnel (CAPES) for financial support.

**Conflicts of Interest:** The authors declare no conflict of interest.

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
