# Peer review of "Fed-Batch Sucrose Crystallization Model for the B Massecuite Vacuum Pan, Solution by Deterministic and Heuristic Methods"

_processes, doi:10.3390/pr8091145_

Round 1

Reviewer 1 Report

what is the final product looks like, any photo for the crystals?

The figure 1 needs to be described with more details.

what is experimental condition for exp. 1 and 2?

The R2 is for fitting witch experiment?

D(4,3) is sued, do author has size distribution?

Three methods all have very good consistence with the experimental date, compare with the derivation of the date, the three methods are all very good. The particle warm method is best method, but what is the application point for this?  

Round 2

Reviewer 1 Report

The authors have addressed most of the comments. I think it is publishable at this stage.

Reviewer 2 Report

The manuscript can be accepted in the present form as the authors have cleared the previous concerns in the revised manuscript.